# Targeting DNA Damage Repair Mechanisms in Pancreas Cancer

**DOI:** 10.3390/cancers13174259

**Published:** 2021-08-24

**Authors:** Lukas Perkhofer, Talia Golan, Pieter-Jan Cuyle, Tamara Matysiak-Budnik, Jean-Luc Van Laethem, Teresa Macarulla, Estelle Cauchin, Alexander Kleger, Alica K. Beutel, Johann Gout, Albrecht Stenzinger, Eric Van Cutsem, Joaquim Bellmunt, Pascal Hammel, Eileen M. O’Reilly, Thomas Seufferlein

**Affiliations:** 1Department of Internal Medicine I, Ulm University Hospital, 89081 Ulm, Germany; lukas.perkhofer@uniklinik-ulm.de (L.P.); alexander.kleger@uni-ulm.de (A.K.); alica.beutel@uniklinik-ulm.de (A.K.B.); johann.gout@uni-ulm.de (J.G.); 2Oncology Institute, Sheba Medical Center, Tel Aviv University, Tel Aviv 52621, Israel; Talia.Golan@sheba.health.gov.il; 3Digestive Oncology Department, Imelda General Hospital, 2820 Bonheiden, Belgium; Pieter-Jan.Cuyle@imelda.be; 4University Hospitals Gasthuisberg Leuven and KU Leuven, 3000 Leuven, Belgium; eric.vancutsem@uzleuven.be; 5IMAD, Department of Gastroenterology and Digestive Oncology, Hôtel Dieu, CHU de Nantes, 44000 Nantes, France; Tamara.matysiakbudnik@chu-nantes.fr (T.M.-B.); estelle.cauchin@chu-nantes.fr (E.C.); 6GI Cancer Unit, Erasme Hospital, Université Libre de Bruxelles, 1070 Brussels, Belgium; JL.VanLaethem@erasme.ulb.ac.be; 7Vall d’Hebrón University Hospital and Vall d’Hebron Institute of Oncology, 08035 Barcelona, Spain; tmacarulla@vhio.net; 8Institute of Pathology, University Hospital Heidelberg, 69120 Heidelberg, Germany; albrecht.stenzinger@med.uni-heidelberg.de; 9Medical Oncology, University Hospital del Mar, 08003 Barcelona, Spain; jbellmun@bidmc.harvard.edu; 10Beth Israel Deaconess Medical Center, Harvard Medical School, Boston, MA 02215, USA; 11Hôpital Beaujon, 92110 Clichy, France; pascal.hammel@aphp.fr; 12Gastrointestinal Oncology Service, Memorial Sloan Kettering Cancer Center, New York, NY 10065, USA; oreillye@MSKCC.ORG; 13Department of Medicine, David M. Rubenstein Center for Pancreatic Cancer Research, Memorial Sloan Kettering Cancer Center, New York, NY 10065, USA

**Keywords:** DNA damage repair, pancreatic ductal adenocarcinoma, *BRCA1/2*, PARP inhibition, platinum, homologous repair deficiency

## Abstract

**Simple Summary:**

Pancreatic cancer is a devastating malignant disease with a dismal prognosis and limited treatment options. Around 14% of pancreatic cancers harbour mutations in specific genes that are important to ensure appropriate DNA repair after damage, like the *BRCA 1* and *2* genes. Recently, with olaparib a first treatment option for *BRCA 1* and *2* mutated pancreatic cancer was approved. However, there is a relevant proportion of further genes involved in the DNA damage repair beyond *BRCA1* and *2* that might benefit from such tailored therapeutic interventions like olaparib. Unfortunately, due to the lack of specific data, no general recommendations are currently available. Therefore, a representative panel of experts was assembled by the European Society of Digestive Oncology (ESDO) to assess the current knowledge and evaluate the significance to treat pancreatic cancer with mutations in DNA damage repair genes. The data-driven consensus recommendations of the ESDO expert panel aim to provide clinicians guidance for a state-of-the-art management.

**Abstract:**

Impaired DNA damage repair (DDR) is increasingly recognised as a hallmark in pancreatic ductal adenocarcinoma (PDAC). It is estimated that around 14% of human PDACs harbour mutations in genes involved in DDR, including, amongst others, *BRCA1/2*, *PALB2*, *ATM*, *MSH2*, *MSH6* and *MLH1*. Recently, DDR intervention by PARP inhibitor therapy has demonstrated effectiveness in germline *BRCA1/2*-mutated PDAC. Extending this outcome to the significant proportion of human PDACs with somatic or germline mutations in DDR genes beyond *BRCA1/2* might be beneficial, but there is a lack of data, and consequently, no clear recommendations are provided in the field. Therefore, an expert panel was invited by the European Society of Digestive Oncology (ESDO) to assess the current knowledge and significance of DDR as a target in PDAC treatment. The aim of this virtual, international expert meeting was to elaborate a set of consensus recommendations on testing, diagnosis and treatment of PDAC patients with alterations in DDR pathways. Ahead of the meeting, experts completed a 27-question survey evaluating the key issues. The final recommendations herein should aid in facilitating clinical practice decisions on the management of DDR-deficient PDAC.

## 1. Introduction

Pancreatic ductal adenocarcinoma (PDAC) has a devastating prognosis. Accounting for only 3.2% of all new cancer cases in the USA, PDAC emerges as having one of the highest mortality rates, ranking as the third most lethal malignancy [1]. Amongst others, the particular features of PDAC, including lack of clinical symptoms for early diagnosis, high resistance to treatment and a rising incidence, project PDAC to displace colorectal cancer (CRC) as the second most common cause of cancer-related mortality by 2030 [2]. Median overall survival time in advanced PDAC rarely exceeds 1 year, and the relative 5-year survival remains around 10%, all stages included, compared to nearly 65% in CRC [1]. Large-scale genomic analyses have promoted precision oncology aiming at identifying novel targets and designing new drugs to be implemented within the framework of personalised therapy. Molecular analysis has demonstrated up to 63 genetic alterations in a single PDAC assigned to 12 core signalling pathways, with DNA damage control being one of the key pathways [3]. Subsequent analysis allowed subtyping of PDAC according to aberrations of chromosomal structure, permitting prediction of treatment response. Waddell and colleagues defined four PDAC subtypes: (i) stable, (ii) locally rearranged, (iii) scattered and (iv) unstable [4]. Almost 20% of PDAC patients harbour one or more somatic/germline mutations in genes involved in the DNA damage repair (DDR), such as *BRCA1*, *BRCA2*, *PALB2*, *RAD51C*, *RAD51D* and *ATM* in homologous recombination repair (HRR) or *MSH2*, *MSH6* and *MLH1* in mismatch repair (MMR), and leading in up to 14% of all PDAC cases to encompass the so-called ‘unstable subtype’ [4,5,6]. These genes that commonly cluster in inherited PDAC [7] are mostly relevant for normal function of HRR [8]. Alterations in these genes can lead to a homologous recombination repair-deficient (HRD) phenotype within a given tumour. The terms ‘BRCAness’ and ‘HRDness’ are therefore partly used interchangeably, without clearly defined consensus definitions. More precisely, ‘BRCAness’ describes a molecular, histological, clinical and therapeutic (PARP inhibitor, topoisomerase inhibitor and platinum agent sensitivity) mimicry of a germline *BRCA1/2* loss, over and above HRD [9,10]. ‘HRDness’ extends this genetic spectrum to other somatic or germline mutations causing defective HRR, including mutations in non-*BRCA* HRR genes [5,10]. This could also include HRD-related genomic scar signatures related to *BRCA1/2* mutations with loss of heterozygosity, telomeric allelic imbalance [11] or large-scale state transitions [12], although we do not have detailed knowledge on their precise role at this juncture. A significant proportion of human PDACs with either somatic or germline mutations in DDR genes might benefit from targeted therapies [13], as shown for the PARP inhibitor (PARPi) olaparib as maintenance treatment in metastatic, germline mutated *BRCA1/2* (g*BRCA*) PDAC. The phase III POLO trial demonstrated significantly improved progression-free survival (PFS) as compared to placebo (7.4 vs. 3.8 months; HR 0.53, *p* = 0.004) using olaparib as maintenance therapy in this patient population, following objective disease control on platinum-based first-line chemotherapy for at least 16 weeks [14]. However, g*BRCA1/2* mutations comprise only a small proportion of genes that are involved in DDR, whereas other genes are more common; however, data regarding their role in HRD/DDR therapy is lacking [5,6]. Further insight was recently given by the application of various HRD classifiers on the whole genome sequencing dataset of 391 PDAC patients. An HRD signature could be attributed to alterations in *BRCA1/2*, *PALB2*, *RAD51C/D, XRCC2* and a tandem duplicator phenotype. In advanced disease, the HRD signature was predictive for platinum response and survival benefit [15]. Furthermore, therapeutic approaches for DDR gene mutations have been underpinned by the concept of synthetic lethality in the preclinical [16] and clinical settings [17,18].

However, clinical data published so far only support the use of the PARPi olaparib in the therapeutic maintenance setting. This can be complemented by the use of platinum derivatives due to their recognised interaction with DNA bases, which translates to DNA damage [19]. As for most targeted therapies, the evidence of the efficacy of platinum derivatives in patients with germline and somatic *BRCA1/2* mutations is currently mainly based on retrospective analyses. However, lately, it has been shown that biallelic somatic or germline mutations in HRR genes, namely, *ATM*, *BARD1*, *BLM*, *BRCA1*, *BRCA2*, *CHEK2*, *PALB2*, *RAD50* and *RAD51C* have a higher tumour mutational burden and the strongest association with genomic instability compared to wild-type tumours. In line, outcome improvement in these HRD patients was observed when platinum compounds were part of the first-line therapy of advanced PDAC [20,21].

In conclusion, the current developments allowing for molecular stratification of PDAC change the therapeutic landscape but also engage therapeutic uncertainty. Most importantly, accumulating preclinical evidence suggests an extension of DDR interference strategies to other, non-*BRCA* mutated but DDR-defective PDAC. This evidence, however, needs to be considered with caution, as true clinical evidence is lacking. Clear recommendations for routine clinical diagnostic and therapeutic strategies for this relevant PDAC subtype are urgently warranted.

### Aim and Scope

The expert meeting of the European Society of Digestive Oncology (ESDO) focused on DNA damage pathways in PDAC and implications on routine clinical practice. The aim of the meeting was to generate/provide data-driven consensus recommendations to underpin guidance of the clinical management of patients with PDAC and DNA repair deficiency and for testing strategies.

## 2. Materials and Methods

### Composition of the Expert Panel and Recommendations

The expert panel consisted of 11 international experts in oncology from 6 nations (Belgium, France, Germany, Israel, Spain, USA). The participants completed a survey of 27 questions in advance (Table 1). The subsequent ESDO-hosted expert meeting took place online on 22 July 2020. An interim report for assessing the results and recommendations was sent to the participating experts for approval, additional commentary and suggestions, as well as for voting/ranking their Level of Agreement (LoA). The ranking spans of a scale of 1–4, where A means I completely agree, B means I agree with minor reservation, C means I agree with major reservation and D means I disagree. A consensus recommendation was considered/granted when >80% of experts voted to accept the statement completely or with minor reservation. Additionally, the consensus statement was complemented by a compilation of four sequencing panels used by the experts in their daily practice.

## 3. Results

The results of the subsequently listed experts’ voting showed a high concordance for diagnostic strategies to identify patients with DDR mutations and provide therapeutic approaches. The recommendations from the European Society of Digestive Oncology expert panel are summarized in an algorithm, to facilitate clinical practice decisions on management of DNA damage repair-deficient pancreatic ductal adenocarcinoma (Figure 1).

### 3.1. The Incidence of Germline BRCA1/2 Mutations

The distribution of g*BRCA* mutations varies regionally, e.g., due to specific communities and population migration [22]. Thus, local g*BRCA* testing strategies can be influenced by various factors, including regional areas of high incidence, but also due to reimbursement of testing policy. The majority of experts are aware of the g*BRCA1/2* mutation frequency in their respective regions. The average rate of g*BRCA* mutations in patients with PDAC is about 6% (range 2–12%), with a high concordance between European countries and the USA. The proportion of g*BRCA*-mutated PDAC patients in Israel is substantially higher (12%). This is in line with the published literature, though that includes no structured population-based analysis but rather selected PDAC collectives analysed within therapeutic trials or single-centre experiences [23,24,25,26,27,28]. As outlined, regional distribution can be influenced by higher proportions of at-risk populations, e.g., the Ashkenazi population [29,30]. Analysing the biggest reported experience so far, the POLO (Pancreas OLaparib Ongoing) trial provides the most accurate insight in g*BRCA1/2* distribution, demonstrating a ratio of 7.5% mutations in 3315 patients. However, selection bias has to be considered due to the inclusion/exclusion criteria of the trial, given that 19.8% of patients included in the trial had a previously known g*BRCA* mutation. Concerning sex distribution, the POLO trial was only a little in favour of the male sex. In a study following 8140 pedigrees from *BRCA* 1/2 mutation carriers, 351 developed pancreatic cancer, with 203 (58%) male and 148 (42%) female cases [31]. Overall survival was not significantly different between male and female PDAC in this study or in the POLO trial [31].

### 3.2. BRCA1/2 Testing Recommendations

Current society-based guidelines differ in their recommendations regarding genetic testing for g*BRCA* mutations in patients with PDAC. This may partly be due to the fact that some guidelines have not yet been updated according to the rapidly growing body of evidence supporting this recommendation [32]. The most recent National Comprehensive Cancer Network (NCCN) clinical practice guidelines on “Pancreatic adenocarcinoma” version 1.2020 and “Genetic/familial high-risk assessment: Breast, ovarian and pancreatic” version 1.2020 [33] recommend germline testing for any confirmed PDAC patient using comprehensive gene panels for hereditary cancer syndromes. This recommendation is followed by about half of the panel experts. The other half does not routinely perform *BRCA* mutation screening at present, mainly for logistical reasons in their respective country, including a lack of resources for genetic testing and/or reimbursement concerns.

In principle, the panel recommends genetic testing for germline *BRCA1/2* mutations for all advanced PDAC patients fit to undergo any cancer-specific therapy (LoA: A 90%, B 10%).

As the panel is aware of shortcomings in testing capacities and reimbursement by the respective health care systems, the panel agreed on definitive indications for g*BRCA1/2* testing in PDAC, including (i) cases with a positive family history for PDAC, (ii) previous *BRCA*-associated tumours, (iii) young age at the initial diagnosis and (iv) in patients with a good response to platinum therapy (LoA: A 80%, B 20%).

There are data suggesting that up to 42% of pathogenic germline mutations in PDAC patients may be missed by employing a focused testing strategy [34]. In a cohort of 854 sporadic PDAC patients, 33 (3.9%) had deleterious germline mutations, of whom only 15 were g*BRCA1/2* mutations [24]. Thus, prospective collection of data on the relevance and clinical impact of mutations in other non-*BRCA* HRR genes is urgently warranted.

The panel agreed that PDAC patients might benefit from an even broader testing strategy with regard to mutations in genes involved in the HRR pathway (LoA: A 60%, B 40%).

The optimal time point of testing might differ according to the perspective considered: relevance for the patient or relevance for the family (see genetic counselling). Regarding the relevance for patients with PDAC, the majority of participating experts endorsed testing at the time of initial diagnosis of advanced PDAC or in advance of commencing first-line therapy (63.6%). This is due to the expected good response of these tumours to platinum-containing therapies. About one-third of the experts proposed testing prior to second- or third-line therapy. All participants agreed that *BRCA*-mutated PDAC patients should be enrolled into ongoing prospective clinical trials where feasible [35]. A minority of experts propagated an extension of testing to all settings of PDAC, including patients with resectable or borderline resectable PDAC, since it might further enable optimisation of neoadjuvant treatment strategies. Retrospective data suggest an increase in complete pathologic responses after neoadjuvant FOLFIRINOX (5-fluorouracil, leucovorin, irinotecan and oxaliplatin) for *BRCA*-mutated PDAC patients with borderline resectable disease [36], although there are no robust data available yet. Testing for germline mutations also has important implications for the patients’ family and should lead to genetic counselling (see below).

With respect to the material used for *BRCA* analysis, about half of the experts uses both blood and tissue-based *BRCA* testing. One-third uses only blood and one-fifth only tumour tissue. The testing is mainly performed by molecular pathologists and/or geneticists. For analysing the germline only status, the majority (90.9%) of experts utilise next-generation sequencing (NGS) panels to assess the g*BRCA1/2* status covering a broad spectrum of relevant mutations. Limited analysis of only g*BRCA1/2* gene mutations is conducted by 36.4%, but only one (9.1%) expert advocated to rely solely on this analysis.

The experts agreed that a single specific analysis should be used for screening of relatives of an affected patient (LoA: A 22.2%, B 77.8%).

As stated above, mononuclear blood cell testing is sufficient for g*BRCA1/2* testing and is more readily available compared to tissue. The literature demonstrates similar results using mononuclear blood cells or tumour tissue with respect to *BRCA* germline mutations [37]. In the near future, decision making in PDAC may be conducted by liquid biopsy analysis of mutations in genes encoding the HRR pathway, as recently approved for specific PARPi therapy in castration-resistant prostate cancer [38,39].

The panel agreed that a turnaround time of 4 weeks (2–8 weeks) for the results of g*BRCA1/2* testing is acceptable (LoA: A 88.9%, B 11.1%), but this target is not achieved in all countries [37]. However, a shortening of the interval is currently not mandated in view of limited currently available therapies and indications [14].

### 3.3. Genetic Counselling for Patients with a Germline BRCA1/2 Mutation

Familial pancreatic cancer accounts for up to 10% of PDACs. In-depth genetic testing is highly recommended for these patients and family members.

Consequently, the panel agreed that patients with a positive family history or patients with a g*BRCA* mutation result should be referred to a geneticist at primary diagnosis of the PDAC (LoA: A 80%, B 20%).

Patients with a positive family history of cancer, particularly PDAC, should also be referred for genetic counselling independent of the g*BRCA1/2* mutation status (LoA: A 80%, B 10%).

The experts also agreed that obtaining the family history of the patient is an important factor to identify patients with familial pancreatic cancer as defined by >1 affected first-degree relative or *BRCA*-associated cancers (LoA: A 80%, B 20%). However, it has to be highlighted that patients with g*BRCA1/2* mutation often do not have a family history of cancer [24].

Genetic testing of first-degree relatives of PDAC patients is generally recommended, but considering the restrictions discussed above, it appears mandatory in at least the setting of positive family history for cancer or at a young age at the initial diagnosis.

This is in line with NCCN guidelines for pancreatic cancer. In specific situations with a strong suspicion of familial pancreatic cancer, the need for in-depth molecular analysis of the tumour as well as parallel referral to a geneticist are recommended by the panel. Genetic counselling was identified as a potentially limiting factor due to a lack of skilled counsellors and other resources.

### 3.4. Analysing Homologous Recombination Repair Pathways

Multigene next-generation sequencing allows the parallel evaluation of a high number of mutations from a given tumour in a short time. Most participating centres (90.9%) use extended NGS gene panels, summarised in Table 2. Several commercial PDAC germline testing panels are available [6], shown in Appendix A.

The majority of the experts use extended panels to identify mutations in other non-*BRCA1/2*-related HRR genes. This is conducted to elaborate novel therapeutic approaches or to decide on a platinum-based treatment strategy or potentially a PARPi therapy. Some of the experts also recommend such a panel to assess prognosis. However, the majority of the experts determine HRD panels only within the framework of a clinical trial (e.g., ATM-targeted therapy) or in collaborative research projects.

The therapeutic relevance of different mutations in *BRCA1* or *BRCA2* remains poorly defined in many cancers. Only deleterious *BRCA* mutations are likely to have prognostic relevance. For PARP inhibitor effectiveness in *BRCA1/2*-mutated tumours, lineage rather than the mutational origin (germline vs. somatic) or zygosity seem to be relevant [40]. Indeed, *BRCA1/2* biallelic inactivation is not required to confer a relevant vulnerability to PARPi in *BRCA*-associated cancers, as heterozygote *BRCA* patients have shown sensitivity at a level similar to germline carriers [40]. For the majority of experts (63.6%), a specific class 4 or 5 *BRCA1/2* mutation according to ACMG criteria [41] currently does not influence their clinical decision. Four (36.4%) experts consider specific *BRCA1/2* mutations for taking practice decisions.

The experts agreed that upon detection of a *BRCA1/2* mutation, an assessment for pathogenicity should be performed using tools and algorithms designed to distinguish driver mutations from passengers and databases aggregating information about clinical significance, e.g., CHASM, COSMIC, ClinVar, BRCA Exchange or OncoKB (LoA: A 100%).

### 3.5. Treatment of PDAC Patients with BRCA1/2 Mutations

Platinum agents can interrupt DNA transcription and stall replication by crosslinking purine bases in DNA and thereby aggravating DNA damage caused by a limited DNA damage repair capacity [19]. As there are no data available comparing routinely used platinum derivates in a head-to-head manner, it remains unclear whether one platinum derivative might be superior to another. The registry trial Know Your Tumor Program has outlined the value/importance of platinum therapy in patients with germline or somatic *BRCA1/2* or *PALB2* mutations in advanced PDAC. Six patients with platinum-naïve advanced PDAC were compared to a cohort of 32 patients who received platinum. Despite the small numbers, in those patients, the platinum treatment showed a clearly beneficial effect and predictive value (platinum-naïve mOS = 0.71 years vs. platinum-treated mOS = 1.56 years). Furthermore, survival times exceeded the ones of HR-proficient PDAC patients treated with platinum (mOS = 1.45 years). Beyond that, the registry data reflect the prognostic value of *BRCA1/2* or *PALB2* mutations as the survival times are considerably shorter compared to the mOS of 1.13 years in HR-proficient platinum-naïve PDAC patients (n = 113) [42]. In line, platinum treatment in 22 *BRCA1/2*-mutated patients with advanced PDAC significantly prolonged the survival when compared to a similar group of 21 platinum-naïve patients (22 vs. 9 months; *p* < 0.039) [43]. These data are prospectively validated in a recent phase two trial, strengthening the efficacy of first-line therapy with gemcitabine plus cisplatin in 50 advanced g*BRCA1/2* or *PALB2*-mutated PDAC patients (mOS 16.4 months) [44]. FOLFIRINOX may confer an additional benefit, as topoisomerase inhibition has also shown efficacy in DDR-deficient preclinical PDAC models [16].

In the setting of a confirmed pathogenic germline *BRCA* mutation, all experts agree on initiating a platinum-based therapy, preferably (m)FOLFIRINOX (LoA: A 100%). The majority of experts (LoA: A 81.8%) recommend treating a patient with a somatic *BRCA1/2* mutation in their tumour in a similar manner.

A combination of gemcitabine with cisplatin or oxaliplatin can also be considered (LoA: A 100%).

The recently published POLO trial reported a significant prolongation of PFS with a PARPi maintenance treatment compared to a placebo [14]. Although showing a trend for prolonged overall survival (HR 0.83), the follow-up data of the trial could not demonstrate a significant survival benefit in the experimental arm [45]. This raises the question of whether prolonged PFS is a clinically meaningful endpoint. PFS was chosen as the primary endpoint because patients with a g*BRCA* mutation who have received platinum chemotherapy have prolonged survival and receive multiple subsequent therapies, which could affect OS. The experts emphasised that some patients received PARPi subsequent to the trial [14,45]. The majority of experts (LOA = 90.1%) rated PFS as a clinically meaningful endpoint in patients with *BRCA1/2* germline mutations and pancreatic cancer and therefore is a sufficient parameter for clinical use.

Based on the results of the POLO trial, the majority of experts recommend and routinely use olaparib as maintenance therapy in g*BRCA1/2*-mutated PDAC patients if the drug is reimbursed by the respective health care system or is accessible through a medical need program. In the USA, olaparib was approved by the Food and Drug Administration (FDA) for the maintenance therapy of adult metastatic PDAC patients with a deleterious or suspected deleterious g*BRCA* mutation (detected by an FDA-approved test) in the setting in which at least stability (or response) was observed following 16 weeks of platinum-based therapy [46]. Olaparib was approved in the European Union by the European Medicines Agency in 2020 [47]. Of note, platinum-based chemotherapies and PARP inhibitor maintenance provided an equivalent clinical benefit for both g*BRCA1*- and g*BRCA2*-mutated pancreatic cancer patients in these different trials.

### 3.6. Treatment of PDAC Patients with Other Mutations in DDR Genes

Apart from g*BRCA1/2* mutations, there are somatic (and rarely germline) mutations in other HRR genes, such as *ATM*. Despite limited data, most experts (81.8%) stated that they treat patients with PDAC and proven somatic or germline DDR gene mutations such as those with g*BRCA*-mutated PDAC, i.e., with platinum-based chemotherapy, especially FOLFIRINOX. Data from the Know Your Tumor Program identifies three groups of relevant HHR genes (mutation frequencies as stated in the publication): group 1 *BRCA1* (1.8%), *BRCA2* (4.8%) and *PALB2* (0.9%) with clinical evidence of a predictive value; group 2 *ATM* (4.6%), *ATR* (0.4%) and *ATRX* with strong preclinical evidence of a putative predictive value; and group 3, including the MRN complex (*MRE11*, *RAD50* and *NBN* and other downstream effectors) (0.2%) and the Fanconi Anemia core (*CHEK2* (1.0%), *FANCA*, *FANCC/G/L* (1.0%) and *BLM* (0.2%) genes that have preclinical evidence in HRD. Although the individual groups are small, the authors concluded that in advanced PDAC patients, HRDness is potentially predictive for response to platinum-based therapy, but not prognostic, albeit tumours bearing these mutations exhibit a more aggressive tumour phenotype [42] in line with preclinical data [16,48,49]. The individual median overall survival times for treatment with or without platinum therapy were as follows: HRR proficient: 1.45 years (n = 258 patients) vs. 1.13 years (n = 113); group 1: 1.56 years (n = 32) vs. 0.71 years (n = 6); group 2: 2.05 years (n = 15) vs. 0.97 years (n = 7); group 3: 2.39 years (n = 6) vs. 1.10 years (n = 6) [42]. Analysing a group of 50 HRD PDAC patients, germline and somatic mutations in *ATM*, *BARD1*, *BLM*, *BRCA1*, *BRCA2*, *CHEK2*, *PALB2*, *RAD50*, *RAD51C* genes were identified. Progression-free survival was significantly improved when HRD patients were treated with a platinum-containing first-line therapy; this effect was less pronounced for OS. Increased genomic instability and a deleterious outcome were equally observed for somatic and germline HRR mutations [21].

There is a substantial number of variants of unknown significance (VUS) in genes encoding the HRR pathway that continues to expand with the increased use of NGS.

The panel recommends that PDAC patients with germline or somatic mutations in non-*BRCA*-DDR genes in the tumour should be treated with a platinum-containing therapy, preferentially FOLFIRINOX (LoA: A 33.3%, B 66.7%). Whenever possible, these patients should be included in clinical trials.

Checkpoint inhibitors play an increasing role in the treatment of MMR-deficient tumours [50,51]. This raises the question of whether DDR genes mainly involved in HR may also have a potential role in sensitising cancers to immunotherapy. Whole genome sequencing in a discovery (n = 160 PDAC cases) and validation (n = 95 PDAC cases) cohort of resected PDAC revealed distinct somatic mutational signatures. Approximately 12% of cases showed features of genomic instability, characteristic of MMR and double-strand break repair. These were associated with increased transcriptional and immunohistochemical characteristics of antitumour immune activation, including activation of CD8-positive T lymphocytes and overexpression of regulatory molecules such as cytotoxic T-lymphocyte antigen 4 and programmed cell death 1. This corresponded to the augmented frequency of somatic mutations with tumour-specific neoantigens load [52]. Due to the lack of specific data in PDAC, the value of DDR mutations in tumour immunogenicity remains uncertain. Furthermore, in triple-negative breast cancer, the density of tumour infiltrating lymphocytes does not correlate with HRD, at least in early stage cancer patients [53]. In contrast, deleterious DDR mutations in non-small-cell lung carcinoma were associated with improved clinical outcomes in patients treated with PD-(L)1 blockade [54]. Mechanistically, PDAC immunogenicity appears to be associated with/involve autophagy-dependent mechanisms [55].

Given these preliminary and partly controversial data, the panel made no specific recommendation as to whether mutations in DDR genes sensitises PDAC to checkpoint inhibitors (LoA: A 88.9%).

### 3.7. Acquired Resistance to PARP Inhibitor Therapy

Two hallmarks of pancreatic cancer are its high inherent and secondary acquired resistance to chemotherapy. After an initial response to PARP inhibition, timely occurring resistance is quite common [56]. Several studies have shown that most PARPi-resistant *BRCA*-mutated tumours display a hyperactivation of the ATR/CHK1 pathway to maintain genome stability [57,58]. However, the mechanisms of PARP inhibitor resistance in pancreatic cancer are multifaceted and not fully elucidated so far. Preclinical models, for example, linked PARP inhibitor-resistance in ATM-deficient pancreatic tumour cells to the upregulation of drug efflux transporters and detoxification enzymes, and more particularly, an upregulation of alternative-end joining, to counteract the high levels of DNA damage [16].

In line, various approaches might be useful to restore or maintain PARPi sensitivity, such as inhibiting both PARP and ATR/CHK1, as currently evaluated in a phase II trial on a recurrent ovarian cancer cohort (NCT03462342).

### 3.8. Future Directions

Commenting on future directions in PDAC therapy, the experts envisage combinational approaches of multiple DDR inhibitors (54.5%), e.g., ATMi, ATRi or WEE1i. A combination of these inhibitors with chemotherapy (63.6%) or other targeted therapies (54.5%) was also considered. The majority of the experts predicts that checkpoint inhibitors will play a role in the therapy of DDR-deficient PDAC.

Current data on the role of deleterious DDR mutations have largely been obtained in patients with metastatic PDAC. However, also patients in the neoadjuvant or adjuvant setting may benefit from *BRCA* mutation testing. A retrospective case–control analysis of 25 g*BRCA1/2*-mutated PDAC patients compared to 49 HR-proficient patients with surgically resected tumours showed no significant differences in mOS (37.06 vs. 38.77 months, *p* = 0.838) and disease-free survival (DFS) (14.3 vs. 12.0 months, *p* = 0.303). There was a trend towards increased DFS among 10 *BRCA*-mutated cases treated with a neoadjuvant/adjuvant platinum-containing regimens [59]. A retrospective analysis compared 32 patients with resected g*BRCA1/2*- or *PALB2*-mutated PDAC with 64 HR-proficient patients. When not treated with a platinum-containing therapy, survival times were similar between both groups. However, when only patients receiving a platinum-based therapy were considered, mOS was significantly prolonged in 20 HR-deficient compared to 33 HR-proficient patients (47.7 vs. 23.1 months, *p* = 0.032). In addition, perioperative platinum therapy showed a clear mOS benefit for HR-deficient patients when compared to HR-proficient patients (not met at 25.4 months vs. 23.1 months) [60]. The largest analysis thus far is from the Know Your Tumor registry, and showed no survival differences in 63 resected HR-deficient compared to 314 HR-proficient patients irrespectively of the exposure to platinum-based therapy [43]. Due to the lack of prospective trials, the clinical relevance of deleterious DDR mutations in the adjuvant/neoadjuvant situation remains unclear at this time. Current trials are already taking some of these various concepts into account, e.g., NCT04104672, NCT04548752, NCT03983057, NCT04673448, NCT04171700 and NCT04550494.

## 4. Conclusions

The recommendations from an ESDO expert panel considered the totality of the available clinical literature and experience to date in providing these recommendations. While it is unquestionable that these recommendations come with limitations, they provide clinicians guidance for state-of-the-art management of DDR-deficient PDAC.

## Figures and Tables

**Figure 1 cancers-13-04259-f001:**
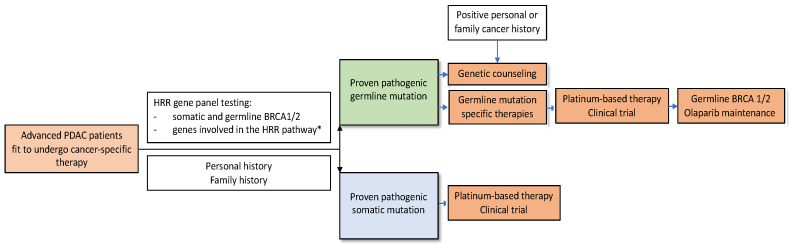
Algorithm to facilitate clinical practice decisions on management of DNA damage repair-deficient pancreatic ductal adenocarcinoma, following the recommendations from the European Society of Digestive Oncology expert panel. HRR, homologous recombination repair; PDAC, pancreatic ductal adenocarcinoma; ***** full panel, see Table 2.

**Table 1 cancers-13-04259-t001:** Overview of the 27-question survey on the role of DNA damage repair in pancreatic ductal adenocarcinoma. DDR, DNA damage repair; HRD, homologous recombination repair-deficient; PDAC, pancreatic ductal adenocarcinoma; PFS, progression-free survival.

#	Question
Q1	Do you know the proportion of g*BRCA1/2* mutation in PDAC in your area/country?
Q2	Do you regularly determine the *BRCA* mutation status in patients with PDAC?
Q2-1	If no, explain why not.
Q3	In which situation do you search for *BRCA* mutation?
Q4	Is your approach different to patients with a suspicion of genetic syndrome and those without any suspicion?
Q4-1	If yes, what is the difference?
Q5	Which material do you use for *BRCA* analysis?
Q6	Who performs the test?
Q7	What is your acceptable/desirable period of waiting for the results?
Q8	In your opinion, does family history play a role in identifying patients with PDAC and g*BRCA1/2* mutation?
Q9	Are patients with g*BRCA1/2* mutation regularly referred to a human geneticist in your country?
Q10	For your testing, do you send the patient to the geneticist before or after checking the results positively?
Q11	Do the panels you use comprise other DDR-related genes?
Q12	Do you determine genomic signatures for HRDness in a given PDAC?
Q12-1	If yes, what is the reason?
Q13	Do you pay attention to differences in the *BRCA* mutational status?
Q13-1	If yes, explain why. Do you think it influences prognosis?
Q14	In the case of a known *BRCA* mutation, is there a preference for a specific chemotherapy combination?
Q15	Is prolongation of PFS compared to placebo a clinically meaningful endpoint for you?
Q16	Do you treat patients with germline *BRCA1/2* mutations with PARP inhibitors as a maintenance treatment?
Q17	Should patients with somatic *BRCA1/2* mutations be treated as patients with g*BRCA1/2* mutations?
Q18	Do somatic (or germline) mutations in other DDR genes have a therapeutic consequence (e.g., *ATM*)?
Q18-1	If yes, what is the proposed treatment?
Q19	Do mutations in DDR genes sensitise genes to checkpoint inhibitors?
Q20	Which developments do you foresee in the area of DNA damage repair deficiency in PDAC without g*BRCA* mutation?
Q21	Do you think we need to consider this *BRCA* status in a (neo)adjuvant setting for a possible application/trial?

**Table 2 cancers-13-04259-t002:** Proposed somatic and germline DDR gene panel—Compilation of participating centres.

*“APC” “ATM” “BAP1” “BARD1” “BLM” “BMPR1A” “BRCA1” “BRCA2” “BRIP1” “CBL” “CDC73” “CDH1” “CDK4” “CDKN1B” “CDKN2A” “CHEK2” “CTNNA1” “DICER1” “EPCAM” “ERCC2” “ERCC3” “ERCC4” “ERCC5” “FAM175A” “FANCA” “FANCC” “FANCD2” “FANCE” “FANCF” “FANCG” “FANCI” “FANCL” “FH” “FLCN” “FLT3” “GREM1” “HOXB13” “IDH1” “IDH2” “LZTR1” “MAP2K1” “MAPK1” “MAX” “MEN1” “MITF” “MLH1” “MRE11” “MSH2” “MSH6” “MUTYH” “NBN” “NF1” “NF2” “PALB2” “PMS2” “POLD1” “POLE” “PRKAR1A” “PTCH1” “PTEN” “RAD50” “RAD51C” “RAD51D” “RAF1” “RB1” “RECQL4” “RNF43” “RUNX1” “SDHA” “SDHAF2” “SDHB” “SDHC” “SDHD” “SLX4” “SMAD3” “SMAD4” “SMARCA4” “SMARCB1” “STK11” “SUFU” “TERT” “TGFBR1” “TGFBR2” “TMEM127” “TP53” “TSC1” “TSC2” “VHL” “WT1” “XRCC2”*

Red Part of several panels used by the experts. Green Part of 2 panels used by the experts.

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
