# Peer review of "Targeting DNA Damage Repair Mechanisms in Pancreas Cancer"

_cancers, 2021, doi:10.3390/cancers13174259_

Round 1
Reviewer 1 Report
The authors have made significant changes to this version. This reviewer is satisfied and recommends acceptance for publication!
Reviewer 2 Report
no
This manuscript is a resubmission of an earlier submission. The following is a list of the peer review reports and author responses from that submission.
Round 1
Reviewer 1 Report
This paper summarizes expert recommendations for BRCA1/2 testing in pancreatic cancers. Pancreatic cancer is perhaps the deadliest cancer particularly because often by the time symptoms develop the cancer is already too advanced for treatment. This is compounded by the fact that the genetic drivers of pancreatic cancers are somewhat of a mystery. This reviewer agrees with the authors that universal recommendation/guidelines for testing genetic markers (in this case BRCA1/2 status) should be established.
However, this reviewer believes that the authors do not go far enough with their recommendations. First, the authors focus on the status of the germline mutations of BRCA1/2 but they bunch BRCA1 and BRCA2 together. As the authors are undoubtedly aware, the functions of BRCA1 and BRCA2 are not identical (see for example a review by Yoshida et al, 2004 Cancer Sci). The authors should make more specific recommendations for BRCA1 and BRCA2, independently.
Second, BRCA1 and BRCA2 haploinsufficiency appears to be enough to drive cellular transformation and immortalization. How are the authors proposing to analyze the results of their proposed tests? Will they distinguish between heterozygous and homozygous mutations? Or is it assumed or known that all germline mutations are heterozygous?
Third, not all BRCA1 and BRCA2 mutations are driver or even pathogenic. Will the authors make the same recommendations for all mutations or will some mutations be considered differently than others? One suggestion would be to use the CHASM algorithm and determine the pathogenic and driver potential of mutations. This could be done using data from COSMIC and/or NCBI. The authors mention that their findings could be deposited into databases such as COSMIC. However, COSMIC already makes predictions on the pathogenicity of cancer mutations (FATHMM scores) which clearly indicates that not all mutations are equal. What will the authors’ study add that COSMIC does not already report?
Fourth, is there a difference between male and female germline mutations? Are female germline mutations more pathogenic than male? This is obviously true for ovarian and breast cancer but is it true for pancreatic cancers? At the very least, a paragraph explaining this issue should be included.
Finally, an organizational recommendation. The authors should make a figure (or perhaps a workflow) of all their recommendations in the results section. If this paper will set standards for BRCA1/2 testing in pancreatic cancers, such a figure would go far in increasing the significance and readership of this paper.
Reviewer 2 Report
This is the summary of conference regarding targeting DNA damage repair signaling in pancreas cancer. As authors realized, the genome of PDAC is very instable, which will complicate the outcomes of treatment with any chemotherapeutic agents, especially for those leading to more DNA damage. In terms of cancer treatment, the accumulated DNA damage in cancer cells is not "more and better". The relative genome stability of cancer cells before and after the treatment with DNA damage agents may be a key to cancer cell sensitivity. The introduced DNA damage from chemo-drugs would lead to an enormous amount of DNA damage when working with defected repair mechanisms "inherited" in cancer cells. This may result in a selection for resistance over a certain period of chemotherapy. This reviewer wishes authors to discuss a bit more about the resistance occurred in cancer patients treated with DNA damage agents and how the resistance can be avoided / improved in patients with PDAC.
Reviewer 3 Report
The paper is a conference paper relating to the role of targeting DNA damage repair mechanisms in PDAC. In the paper, the currently known clinical literature is presented as well as the outcome of the discussion of an expert panel, that was invited by the European Society of Digestive Oncology (ESDO) to assess the current knowledge and significance of DDR as a target in PDAC treatment. This paper is a very useful resource when considering the importance of DDR mutations for future treatment strategies. Unfortunately, the currently available data and experiences do not yet allow to give clear recommendations or directions. A follow up paper in a few years time would be very welcome. I have no criticisms on the content of the paper.
